

# Morphological characterization and staging of bumble bee pupae

Li Tian and Heather M. Hines

Department of Biology, Pennsylvania State University, University Park, PA, USA

## ABSTRACT

Bumble bees (Hymenoptera: Apidae, *Bombus*) are important pollinators and models for studying mechanisms underlying developmental plasticity, such as factors influencing size, immunity, and social behaviors. Research on such processes, as well as expanding use of gene-manipulation and gene expression technologies, requires a detailed understanding of how these bees develop. Developmental research often uses time-staging of pupae, however dramatic size differences in these bees can generate variation in developmental timing. To study developmental mechanisms in bumble bees, appropriate staging of developing bees using morphology is necessary. In this study, we describe morphological changes across development in several bumble bee species and use this to establish morphology-based staging criteria, establishing 20 distinct illustrated stages. These criteria, defined largely by eye and cuticle pigmentation patterns, are generalizable across members of the subgenus *Pyrobombus*, and can be used as a framework for study of other bumble bee subgenera. We examine the effects of temperature, caste, size, and species on pupal development, revealing that pupal duration shifts with each of these factors, confirming the importance of staging pupae based on morphology rather than age and the need for standardizing sampling.

## INTRODUCTION

The primitively eusocial bumble bees are important pollinators and fascinating models for studying developmental plasticity and adaptive variation. Bumble bees contain social castes that engage in division of labor among queens and workers. Social individuals in the nest can differ dramatically in body size, with queens being substantially and discretely larger than workers, and workers exhibiting dramatic plasticity in size, including up to 10-fold size variation (*Couvillon et al., 2010*). Individual size and related social function are influenced by sensitive periods in development to various environmental factors, such as nutrition, temperature and pheromones (*Groh, Tautz & Rössler, 2004*; *Plowright & Jay, 1968*; *Sakagami, Akahira & Zucchi, 1967*; *Tautz et al., 2003*). Deciphering genetic and developmental processes behind these phenotypes remains an active area of inquiry (*Cnaani, Robinson & Hefetz, 2000*; *Couvillon & Dornhaus, 2009*; *Li et al., 2018*; *Pereboom et al., 2005*; *Tasei & Aupinel, 2008*). New genetic technologies increasingly enable the targeting of genes and their functions in non-model organisms.

Corresponding author
Heather M. Hines, hmh19@psu.edu

In bumble bees this expands the potential for uncovering the genetic mechanisms underlying a variety of evolutionarily interesting traits, such as traits related to caste differentiation (*Price et al., 2018*), foraging and social behavior, thermoregulation, diapause (*Amsalem et al., 2015*), coloration (*Rapti, Duennes & Cameron, 2014*), and pollination-relevant morphologies (*Medved, Huang & Popadić, 2014*). The search for gene function has led to the increasing need to assess and modify genes during development. For example, RNAi methods are applied throughout developmental stages to characterize gene function in the bees (*Brutscher & Flenniken, 2015*; *Cridge et al., 2017*; *Hu et al., 2010*; *Medved, Huang & Popadić, 2014*) and many other insects (*Bellés, 2010*; *Gu & Knipple, 2013*), and gene expression studies involve standardization of samples to specific developmental stages (*Chen et al., 2012*; *Collins et al., 2017*; *Jones, Wcislo & Robinson, 2015*; *Pereboom et al., 2005*).

Comparative study of these traits requires a detailed understanding of developmental processes. Larval development has been examined fairly closely in bumble bees. Four larval instars have been determined in *Bombus terrestris* and *B. impatiens* based on frequency distributions of head capsule widths for queen and worker larvae (*Cnaani et al., 1997*; *Cnaani, Schmid-Hempel & Schmidt, 2002*). Finer staging of larval stages has involved categorizing larval instars by weight group (*Cnaani, Robinson & Hefetz, 2000*) and larval duration of different castes has been determined in several species (*Cnaani, Robinson & Hefetz, 2000*; *Cnaani, Schmid-Hempel & Schmidt, 2002*; *Sakagami, Akahira & Zucchi, 1967*). The development of bumble bee pupae is less understood. In general, pupation in bumble bees is similar to other bees (*Michener, 2000*) and many holometabolous insects (*Gillott, 1991*). It starts when a pre-pupa sheds the larval cuticle, a process known as larva-pupal ecdysis. The pupa then undergoes apolysis, in which the newly developed adult epidermis begins to separate from the pupal cuticle, marking the beginning of the pharate adult stage (*Elias-Neto, Soares & Bitondi, 2009*). The end of the pupa stage is marked by molting of pupal cuticle, known as pupal-adult ecdysis, followed by the eclosion of the young adult, termed the callow, from the cocoon. Unlike in larvae, detailed descriptions of the changes during pupal development are lacking.

Many developmental studies on pupae, such as those in *Drosophila*, tend to stage pupae by age (*Berry & Baehrecke, 2007*; *Stern, 1998*; *White et al., 1999*). Age-staging has been applied in a few studies on bumble bees (*Bortolotti, Duchateau & Sbrenna, 2001*; *Dean, 2016*; *Ribeiro, 1994*); however, given that developmental timing may vary by size, caste and environmental conditions (*Cnaani et al., 1997*; *Sakagami, Akahira & Zucchi, 1967*; *Sutcliffe & Plowright, 1990*), morphology-based staging criteria is likely to be a more reliable way to establish developmentally homologous stages in these bees. Seven generalized pupal stages based on pigmentation of the thorax and the compound eyes are recognized in the honey bee *Apis mellifera* (*Michelette & Soares, 1993*; *Rembold, Kremer & Ulrich, 1980*). These staging criteria have been extended to studies on other bee species, with little or no modifications (*Cardoso-Júnior et al., 2017*; *Hartfelder & Rembold, 1991*; *Thorp, 1969*; *Cruz-Landim & Mello, 1967*; *Daly, 1966*). In only a few bee species, including *A. mellifera* and a stingless bee *Melipona scutellaris*, are keys and images available for pupal stages (*Cardoso-Júnior et al., 2017*; *Daly, 1966*;

*Elias-Neto, Soares & Bitondi, 2009*). Thus far, attempts to separate bumble bee pupa by morphology has involved either using a modification of the same six stages for honey bees or a more simplified model (*Dong et al., 2017*; *Li et al., 2010*; *Mänd et al., 2005*). Morphological keys from other species like honey bees are not enough to represent fine increments of pupal development in bumble bees needed for developmental genetic work.

In the present study, we provide a detailed profile and illustrated key of morphological changes during bumble bee pupal development as a reference for future developmental research. In addition to commonly used traits like the compound eyes and thoracic color, we document slight changes in morphological structure in various body parts including face, appendages and ommatidia. Our key includes 20 distinct and maximally informative stages covering the full duration and regular increments of pupal development, built from comparisons among three bumble bee species of the bumble bee subgenus *Pyrobombus*.

As part of this assessment, we also test how environmental and physiological factors affect pupal duration and influence the length of individual stages. The pupal phase comprises a third of the duration of the immature stage of a bumble bee (*Cnaani, Robinson & Hefetz, 2000*; *Cnaani, Schmid-Hempel & Schmidt, 2002*), but can vary in duration due to multiple factors. Pupal duration may vary by caste, as *B. terrestris* and *B. impatiens* queen pupae have been shown to develop for 4 days longer than workers (*Cnaani, Robinson & Hefetz, 2000*; *Cnaani, Schmid-Hempel & Schmidt, 2002*). Body size is also likely to affect pupal duration as larger bees were observed to take longer to develop than smaller bees in *B. atratus* (*Sakagami, Akahira & Zucchi, 1967*) and *B. terricola* (*Sutcliffe & Plowright, 1990*). Nutrition can also play a role, as pupae from colonies with more access to pollen have longer pupal duration (*Sutcliffe & Plowright, 1990*). A negative correlation between temperature and pupal period has been implied in bumble bees, as reduced temperature decreased colony growth rate in *B. impatiens* and *B. affinis* (*Vogt, 1986b*). Similarly, *Cartar & Dill (1991)* observed that cooling cocoons of *B. occidentalis* and *B. melanopygus* for a short period delayed median adult eclosion time. To add to our understanding of developmental plasticity in pupal duration, and better inform the effects of environmental factors on pupal staging, we specifically test the effect of caste, sex, body size, temperature, and species on pupal duration.

## MATERIALS AND METHODS
### Bumble bees
In this study, we examined and compared pupal as well as early adult development in both commercial and wild reared colonies of the eastern Nearctic bumble bee *B. impatiens* Cresson and in two other wild-reared *Pyrobombus* species from the western Nearctic, *B. melanopygus* Nylander and *B. vosnesenskii* Radoszkowski. Commercial colonies of *B. impatiens* were obtained from Koppert Biological Systems (Howell, MI, USA) and maintained prior to study under laboratory conditions (25 °C, 50% RH, total darkness). Wild bumble bee colonies were reared from queens collected from State College,

Pennsylvania for *B. impatiens* and from Southwest Oregon and Northern California for *B. melanopygus* and *B. vosnesenskii*. These colonies were maintained in plastic boxes in the incubator (29 °C, 60% RH, total darkness). All colonies were fed with sugar water containing a preservative (Koppert Biological Systems, Howell, MI, USA) and the same source of fresh honey bee-collected pollen (Swarmbustin' Honey, West Grove, PA, USA).

## Pupa collection

Newly formed cocoon clusters were removed from colonies and a small opening was cut in the cocoons to inspect the stage of brood. Only cocoons containing prepupae or spinning last instar larvae were retained for pupal staging and duration analysis. These were placed on a plastic weigh boat in the incubator (32 °C, 60% RH, total darkness) and were checked every 4 h for pupal ecdysis, as prepupae could not be removed from cocoons without problems with ecdysis. Pupae are easily distinguished from the preceding, larva-like pre-pupal stage by their resemblance to the adult shape, which can be seen from the opening of the cocoon (Fig. 1). Newly molted pupae were immediately removed from their cocoons and placed individually in a small (25 mm) weigh boat (VWR, Radnor, PA, USA) in the incubator (32 or 29 °C depending on the experiment, 60% RH, total darkness).

## Pupal staging and photography

Pupae collected by the above described method were examined under the microscope (25 °C, 50% RH) every 4–6 h until adult ecdysis occurred. Morphological observations of different body parts, including the compound eyes (referred to hereafter as CE), face, legs, mesosoma, and metasomal segments, were documented across development to establish stages and their duration. Developmental staging was based on observations of 48 workers and seven males for *B. impatiens,* collected from five Koppert colonies, three workers and 20 males for *B. vosnesenskii*, collected from two colonies, and nine workers and 11 males for *B. melanopygus*, collected from five colonies. Defined stages were those that were the least variable and thus most informative for staging within and across species. The duration of each pupal stage was documented for *B. impatiens* workers.

Whole pupae as well as specific characteristics used as staging criteria were imaged. High resolution images of whole pupae were taken at each pupal stage using image stacking with a Canon 7D MarkII digital DSLR camera and Zeren Stacker software. Morphological criteria used for staging pupae were imaged with a Nikon Coolpix 4,500 digital camera attached to an Olympus SZ61 microscope using an eye piece adaptor (Optem 257014).

## Examining setal pigmentation during pupal and early adult stages

To visualize temporal patterns of setal pigmentation during pupal stages, overlying pupal cuticle was removed from metasomal terga in Phosphate-buffered saline (PBS) to reveal the developing setae underneath. Images were taken immediately after dissection. Temporal patterns of setal pigmentation post-eclosion from the cocoon, otherwise referred

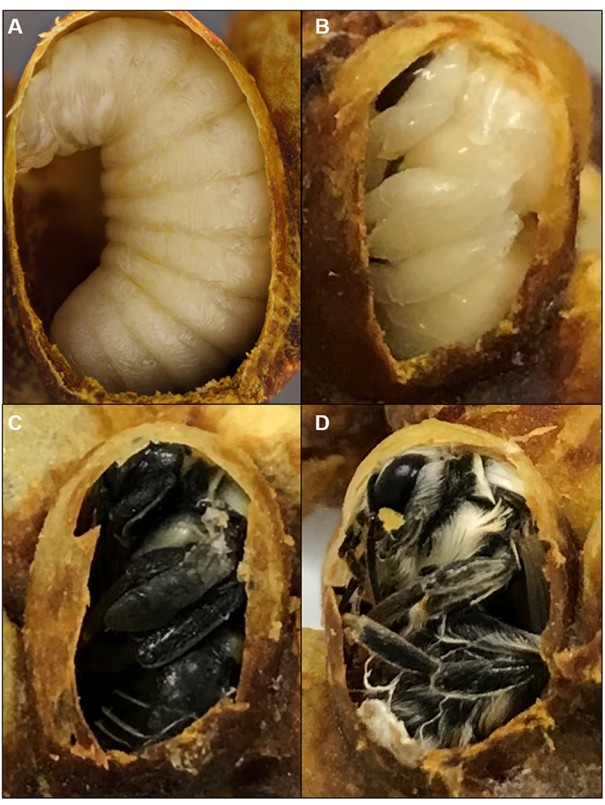

**Figure 1 Cocoons of *B. impatiens* at different stages.** Stages include a prepupa (A), pupa (B) and (C) and a newly eclosed adult bee (D).

to as the callow stage, were visualized for three species—*B. impatiens*, *B. vosnesenskii*, and *B. melanopygus* (red form) by imaging whole bees for different time periods post-eclosion (0, 6, 12, 24, and 48 h). To obtain callows, cocoons containing late stage pupae were kept in the incubator and checked every 4 h for adult eclosion. Newly emerged adults were isolated immediately, reared in a plastic box provided with sugar and pollen, and fresh-frozen (color does not change after freezing) at the appropriate stage.

## Measuring temporal patterns of body melanization

The most variable feature across morphological stages of pupal development in bumble bees is pigmentation. To quantify the temporal pattern of pigmentation for each body part, newly molted pupae of *B. impatiens* were reared at 32 °C and 60% RH and imaged every 12 h until adult ecdysis. Images were transformed to gray scale in Image J and gray values were measured for different body parts, including the CE, head, mesosoma, legs, antennae, and metasoma, for each time point. In gray scale images, the gray value ranges from 0 (white) to 255 (black). To represent the increase of body pigmentation we calculate relative darkness as gray value of area tested/255 X 100%. One image of a whole pupa was used for each time point. We also estimated overall percent melanic body pigmentation by combining scores across all body parts except for the eye, which involves non-melanin pigmentation.

## Measuring pupal duration

Previous studies tended to consider pupation as the period between cocoon formation and eclosion from the cocoon (*Cnaani, Robinson & Hefetz, 2000*; *Sutcliffe & Plowright, 1990*), which thus includes non-pupal stages of spinning larval and prepupal stages, and newly molted adult bees (Fig. 1). We examine the effects of sex, caste, temperature, and species on total pupal duration through observation of pupa reared under controlled environmental conditions outside the cocoon. We are thus able to demarcate pupal duration more accurately as the time between larva-pupa ecdysis (molting of larval cuticle) and pupa-adult ecdysis (molting of pupal cuticle). To measure pupal duration, newly molted pupae collected as above (accurate within 4 h) were left undisturbed in the incubator until the last pupal stage. At this stage, we checked the bees every 2–4 h to document the end of pupation, marked by the pupal cuticle being pushed to the posterior tip of the bee (*Jay, 1965*). Analysis of the effects of body size, caste, sex, and temperature on developmental duration were performed on *B. impatiens*, with individuals drawn from four Koppert ($n = 83$ individuals) and one wild queen-started colony ($n = 24$ individuals). Inter-tegular span (ITS), a common measure of bumble bee body size (*Cane, 1987*; *Hagen & Dupont, 2013*), was measured using a microscope eyepiece ruler (Olympus, Shinjuku, Tokyo, Japan). To avoid potential fluctuations of body size during pupal development, pupal sizes were determined by measuring ITS on newly eclosed adults once the adult ecdysis was complete. We designated three size groups for workers and males: small (ITS < 3.5 mm), medium (3.5 mm $\leq$ ITS < 4.0 mm), and large (ITS $\geq$ 4.0 mm). This grouping was chosen to represent the most meaningful by-eye groupings of typical bee sizes (intermediate) as opposed to those that appear atypically large or small. It matches well with size-based task groups in bumble bee workers recognized by *Goulson et al. (2002)* using *B. terrestris*, where small workers with ITS < 3.5 perform almost exclusively in-nest tasks, large workers with ITS > 4.9 are mostly foragers, and medium sized workers perform both tasks. Because natural bumble bee colonies maintain a nest temperature between 29 and 32 °C (*Heinrich, 1972*; *Pomeroy & Plowright, 1980*; *Vogt, 1986a*), we examined pupal duration for brood reared at 29 and 32 °C to represent the lowest and highest extremes for brood development. Measurements of pupal duration for *B. impatiens* at these temperatures included small worker ($n = 18$ for 32 °C, $n = 7$ for 29 °C), medium worker ($n = 20$ for 32 °C, $n = 6$ for 29 °C), large worker ($n = 15$ for 32 °C, $n = 6$ for 29 °C), small male ($n = 7$ for 32 °C, $n = 3$ for 29 °C), medium male ($n = 13$ for 32 °C), and queen ($n = 8$ for 32 °C, $n = 8$ for 29 °C) treatment groups. A Mann–Whitney test was performed to test temperature effects on pupal duration. To test if there are differences across phenotypes, a one-way ANOVA followed by Tukey's HSD All-Pairwise comparisons were performed for the two temperature treatments separately. To compare between species, we measured pupal duration of small males (ITS < 3.5 mm) of *B. vosnesenskii* (one colony) and *B. impatiens* at both 32 and 29 °C (for *B. vosnesenskii*, $n = 8$ for 32 °C, $n = 3$ for 29 °C) and analyzed the data using a Mann–Whitney test.
## RESULTS

### Structures unique to pupae

Although mostly simply outlining the adult shape, the bumble bee pupal cuticle exhibits a few characteristics not shared with the adult stage. One obvious feature is the metasomal spicules, a row of small spines near the apical margin of metasomal tergites (Fig. 2A). In *B. impatiens*, the spicules are present at metasomal segments two through five (female) or six (for male). These spicules are present throughout pupal development and are shed along with pupal cuticle during pupal-adult ecdysis without replacement in the adult. These structures are absent in pupae of honey bees (Fig. 2B). Another feature is the presence of inner apical protuberances on the coxae and trochanters of the legs (Figs. 2C and 2D). These long protuberances house long setae arising in these segments (Fig. 2E) and appear to act as a brace for the developing legs (Fig. 2C). Honey bee pupae have the same structures at the same location. The coxal and trochanteral protuberances are most distinguishable at the early-mid pupal stage, but collapse and become indistinguishable before pupal-adult ecdysis (Figs. 2F–2H). Similar-shaped structures were also observed on tibial segments, but are retained into adulthood as tibial spines (*Michener, 1954*).

### Pupal stages

We describe 20 pupal stages as well as two early post-ecdysis stages that are applicable to all three studied species and across sexes and castes. Key features for each stage are annotated using illustrations or photos (Figs. 3–5). The defined features primarily involve shifts in eye pigmentation in early stages (Fig. 3), followed by shifts in pigmentation of the developing adult cuticle of the main body parts (Fig. 5). Two distinct cuticle pigmentation processes were observed: a "peppering" process, in which the white cuticle acquires a grayish tinge, often starting as fine gray dots before darkening to black, and a "tanning" process, in which cuticle first turns a more even orange-brown color before turning black. Cuticle pigmentation is later followed by pigmentation of the adult setae, which continues until early adult (callow) stages (Fig. 6) Characteristics differing from *B. impatiens* in other examined species (*B. vosnesenskii, B. melanopygus*), or by caste or sex, are noted. Simplified morphological criteria for each stage are also provided in Table 1. A photographic guide of whole pupae of each stage featuring *B. impatiens* males is presented in Fig. 7.

#### P0, P1, P2: The white-eyed pupa

The white eyed pupa includes three distinct pupal stages, all of which display unpigmented (white) CE and a white body.

P0: The newly molted pupa (stage P0) has a head and thorax resembling the shape of an adult and an abdomen resembling larvae as it bears 10–11 similarly-sized segments (Fig. 3A). Abdominal segments 1, 8, 9, and 10 begin to retract, resulting in visible abdominal twitching (Video S1). The first abdominal segment gradually merges with the thorax to form the propodeum. The last three (8th, 9th, and 10th) abdominal segments telescope into the preceding segments to form the male and female terminalia. At this

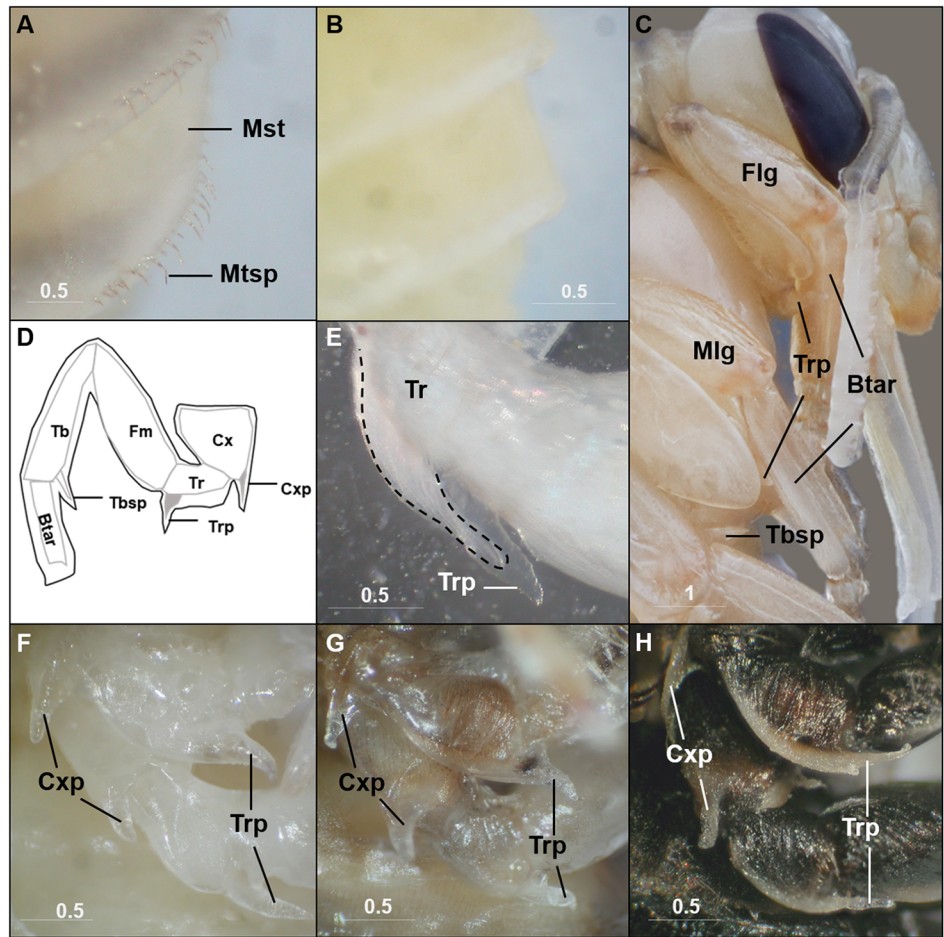

**Figure 2 Characteristics unique to pupal stages.** Metasomal tergal spicules are present in bumble bee pupae (A), but absent in honey bee pupae (B). (C) The trochanteral protuberance (Trp) braces the basitarus (Btar) of the foreleg (Flg) and midleg (Mlg), suggesting a function as a leg hook that maintains proper folding of pupal legs. The tibial spine (Tbsp) of the hind leg is also shown. (D) A schematic illustration of the bumble bee pupal leg showing coxal (Cxp) and trochanteral (Trp) protuberances and tibial (Tbsp) spines. Black lines, pupal cuticle; gray lines, developing adult cuticle; gray shades, developing setae. Illustration is modified from *Michener (1954)*. (E) Trochanteral protuberance with developing adult hair bundle (broken line) encased in pupal cuticle. (F–H), Coxal and trochanteral protuberances of mid and hind legs of early, mid, and late-stage pupae, respectively. The tibia and basitarsus are removed to reveal the protuberances. Note the degeneration of these protuberances in the late pupa (H). Mst, Metasomal tergite; Mtsp, Metasomal tergal spicules; Cx, coxa; Tr, trochanter; Fm, Femur; Tb, tibia; Btar, basitarsus. Scale bar unit: mm.

stage the female and male are readily separated by the sting and male genitalia at the end of abdomen. This stage is the shortest, lasting ~5 h.

P1: This stage begins when retraction of abdominal segments 1, 8, 9, and 10 is completed, and thus abdominal twitching ceases (Video S1; Fig. 3A). This leaves six visible female and seven visible male metasomal segments (Fig. 3A). Leg basitarsi are transparent and cylindrical (Fig. 3B).

P2: Basitarsi become flattened and opaque (Fig. 3B; Video S1). This feature is most obvious in female hindlegs, as it generates the distinct shape of the basitarsal pollen press.

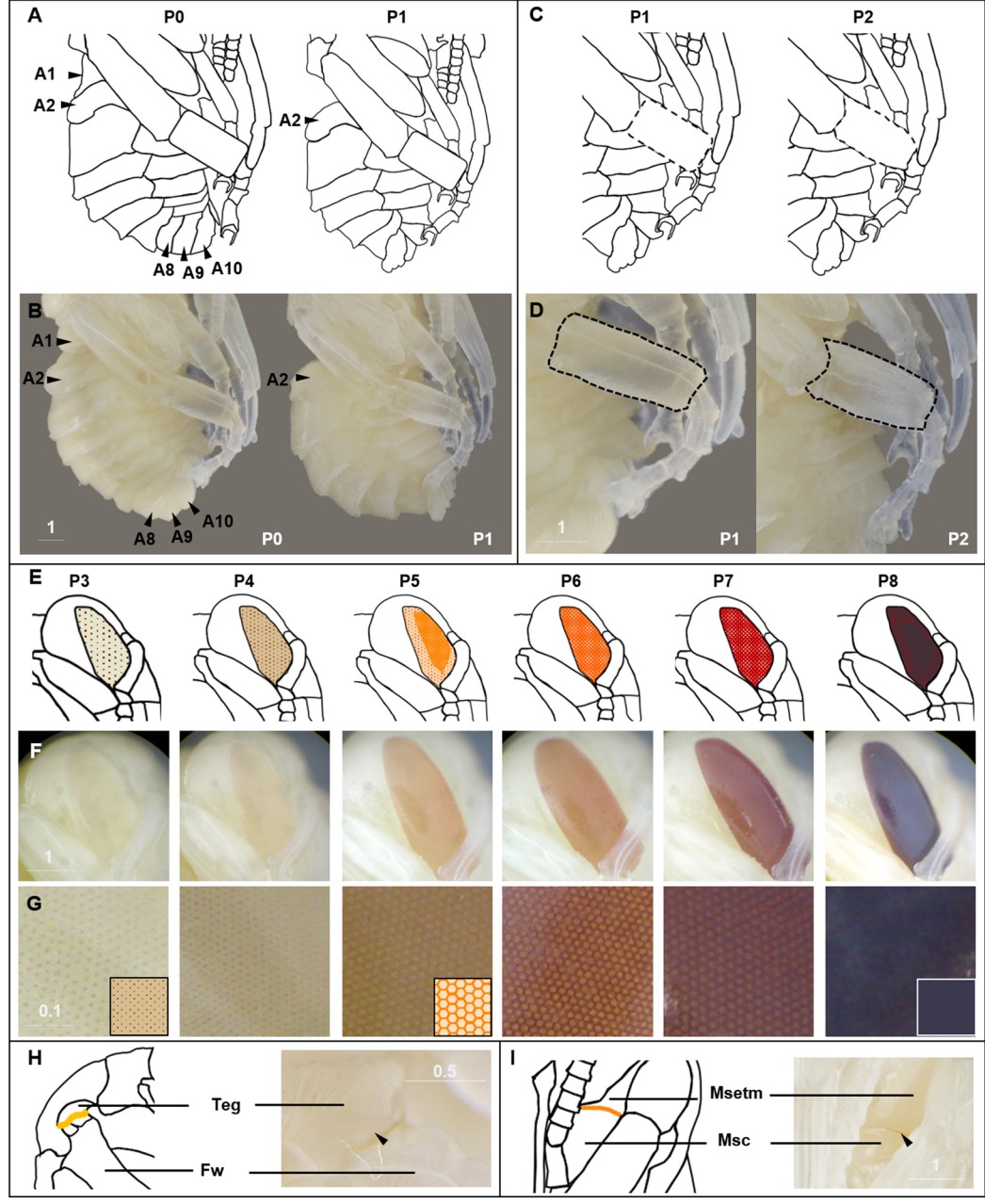

**Figure 3 Morphological criteria marking transitions to P0–P8.** (A & B) Illustration and photo for transition from P0 to P1. Note the retraction of A1 and A8–A10. (C & D) Illustration and photo for changes in shapes of basitarsi of the hind leg (broken lines) of *B. impatiens* workers in the transition from P1 to P2. (E–G) Illustration (E) and photo (F) for color shift of the CE from P3 to P8. (G) shows zoomed view of the CE revealing appearance of ommatidial units at different stages, including dotted (P3 and P4), then hexagonal (P5–P7) and finally, a filled pattern (P8-eclosion). (H) and (I) Black triangles indicate a slightly tanned suture at forewing-tegula junction (H) at P3, and between the mesepisternum and mesocoxa (I) at P7. Teg, tegula; Fw, forewing; Msetm, mesepisternum; Msc, meso-coxa. Scale bar unit: mm.

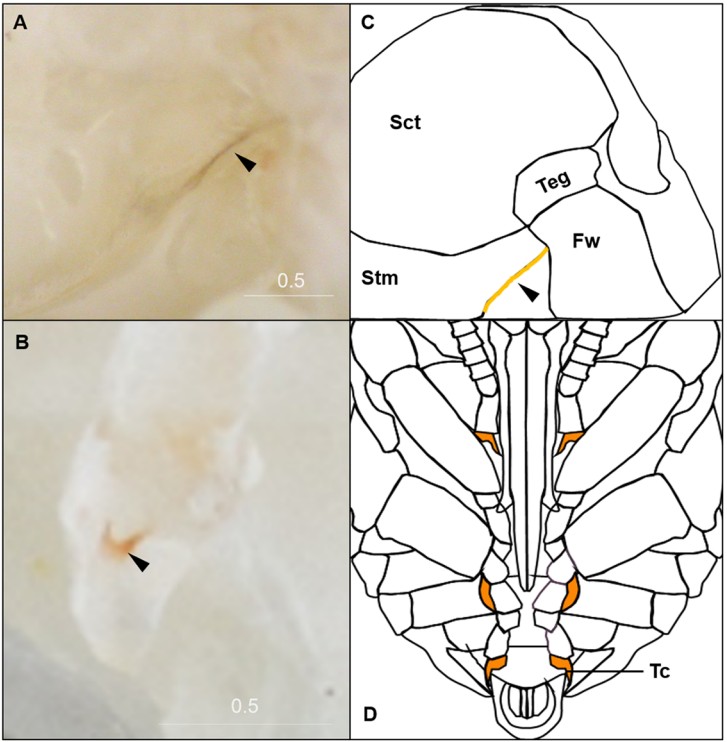

**Figure 4 Morphological criteria marking the transition to P9.** (A & B) The black triangle indicates the tanned suture in the lateral posterior thorax. (C & D) The black triangle indicates a slightly tanned tarsal claw. Sct, scutum; Stm, scutellum; Tc, Tarsal claw. Scale bar unit: mm.

### P3: Yellow-eyed pupa

P3 begins with the first eye pigmentation. The color of the CE is light yellow, different from the surrounding white cuticle. At this time, microscopic light orange-brown dots appear as an ordered array across the eye (Fig. 3C). Often the border between the forewing and tegula becomes slightly tanned, appearing as an orange line (Fig. 3D). This feature is primarily seen in *B. impatiens* females, but not in males or other species.

### P4: Peach-eyed pupa

Ommatidia still appear as dots, but the eye takes on a peach color likely due to intensification of pigments in these dots (Fig. 3C).

### P5: Orange-eyed pupa

This stage starts with the beginning of the transformation of the ommatidia from dotted to hexagonal. The CE now appears orange. Hexagonal transformation starts at the inner margin of the CE and ends when all of the eye contains hexagonal shapes except for the outer CE margin (Fig. 3C).

### P6: Red-eyed pupa

All of the ommatidia are covered with hexagons, including those at the outer margin, resulting in a bright red appearance (Fig. 3C).

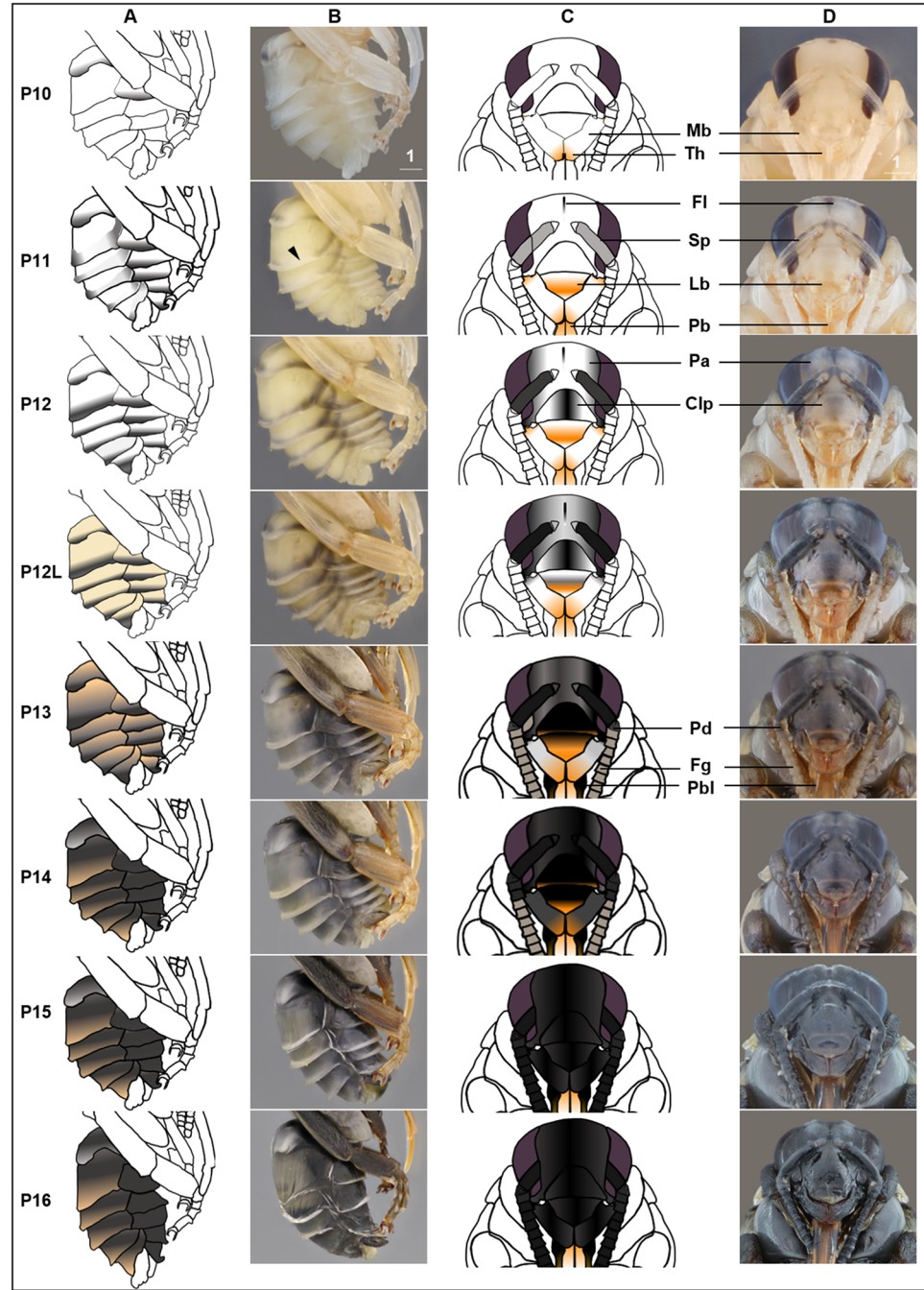

**Figure 5 Morphological criteria marking the transition to P10–P16.** (A & B) Photo and schematic illustration of melanization patterns of abdominal cuticle. Melanization first appears as peppered stripes on the apical margin of each metasomal tergite/sternite disc (P10–P12L), then expands anteriorly to cover the whole disc (P13–P16). The orange tinge at P12L/P13 demonstrates the beginning of tanning of the cuticle. Orange at P14–P16 indicates partial pigmentation of black setae, while the white in T2 (P14–P16) indicates the white color of the future yellow hairs, as the underlying cuticle at these stages is mostly or all black. (C & D) Photo and schematic illustration of pigmentation on the head region including the face, mouthparts and antennae. Mb, mandible; Th, teeth; Fl, frontal line; Sp, Scape; Lb, labrum; Pb, proboscis; Pa, paraocular area; Clp, clypeus; Pd, pedicle; Fg, flagellomeres; Pbl, lateral-basal proboscis. Black triangle: broken melanization stripes at stage P11. Scale bar unit: mm.

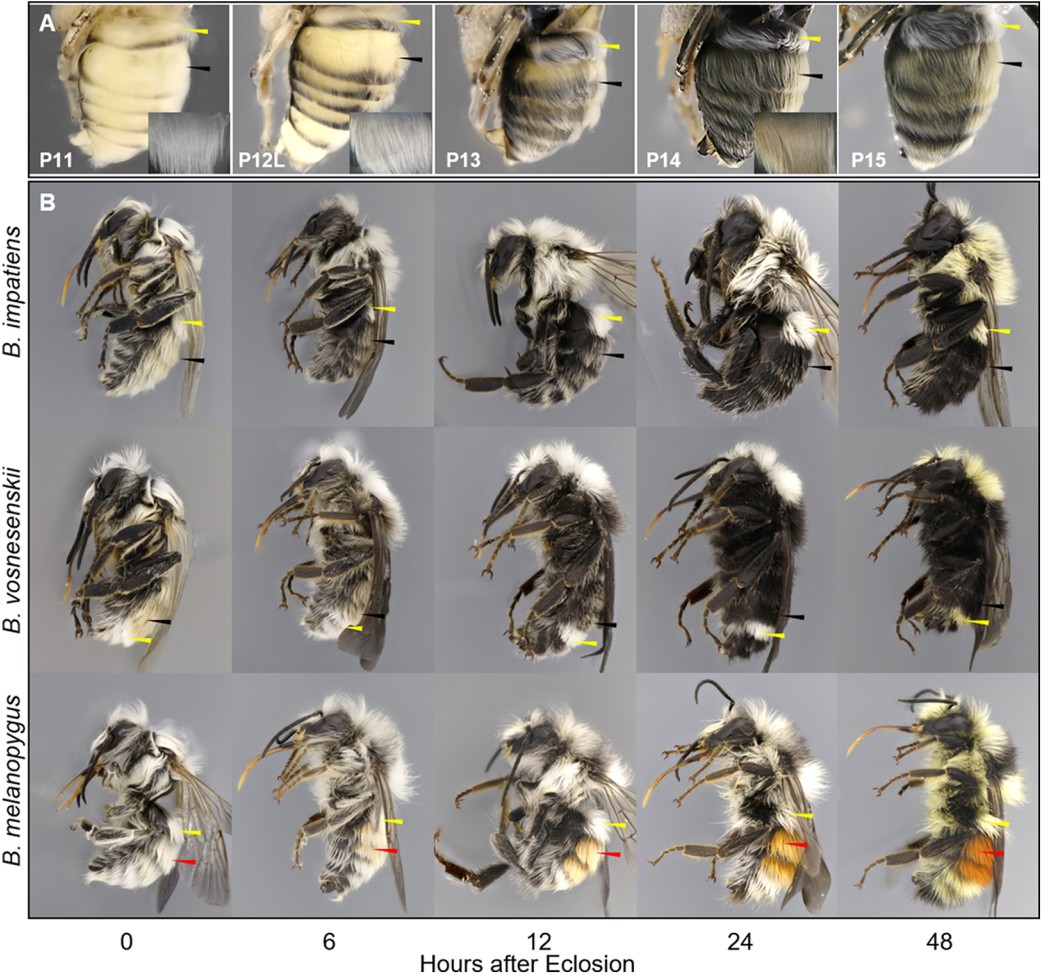

**Figure 6 Temporal changes in setal color in pupal (A) and callow (B) stages.** For (A) metasomal setal color development at late pupal stages of *B. impatiens* visible upon removal of the pupal cuticle. For (B) prospective yellow, black, and red hairs are indicated by yellow, black, and red arrows, respectively. Note the shift in setal color as the callow gets older, as well as the difference between yellow, black, and red setae in the timing of pigmentation. Yellow setae appear to be white until 24–48 h after eclosion, when black and red hairs are nearly fully pigmented.

### P7/P7L: Maroon to reddish-brown-eyed pupa

The CE becomes maroon and progressively darkens to reddish-brown through deepening of the pigment in ommatidial hexagons (Fig. 3C). Being based on eye color, this stage can be difficult to distinguish from P6 at early stages. In *B. impatiens*, another feature marking the beginning of P7 is the tanning of the border between mesocoxa and mesoepisterum, which appears as two orange lines on each side of the mesothorax in the ventral view (Fig. 3E). This feature is absent in *B. vosnesenskii* and *B. melanopygus*. P7 is one of the longest stages, spanning a period of more than a day. This stage can be divided into two stages (P7, P7L) by distinguishing maroon (P7) from reddish-brown eyes (P7L) (Fig. 7).

**Table 1 Generalized staging criteria for pupal development of *B. impatiens*.**

| Stage | Staging criteria | |
|---|---|---|
| | **Primary** | **Secondary** |
| P0 | White eyed, >7 metasomal segments | First abdominal segment not retracted |
| P1 | White eyed, six or seven metasomal segments | Leg segment cylindrical and transparent |
| P2 | White-eyed, Leg segments flat, and opaque | |
| P3 | Yellow-eyed | Tanned suture on tegula/forewing junction* |
| P4 | Peach-eyed | Dotted ommatidia |
| P5 | Orange-eyed | Both hexagonal and dotted ommatidia |
| P6 | Red-eyed | Hexagonal ommatidia |
| P7/7L | Maroon-Brown-eyed | Tanned meso-coxa/mesepisternum suture* |
| P8 | Brown-eyed | Hexagons at eye center filled |
| P9 | Red tinge on mesosoma | Tanned suture on corner of scutellum |
| P10 | Black medial strip on A2 | |
| P11 | Broken apical metasomal stripes | Peppering at frontal line |
| P12/12L | Complete apical metasomal stripes | Gray antennal scape |
| | | Black setae star to turn orange |
| P13/13L | Blackening of lateral-basal proboscis | Black antennal scape, tergites grayish-orange |
| P14 | Abdomen fully black | Flagellomere grayish orange |
| P15 | Antennae fully black | Whole body black |
| P16 | Leg twitching | Extended and wrinkled metasoma |

**Note:**
* *B. impatiens* only.

### P8: Dark brown-eyed pupa

Eye color becomes dark brown. The beginning of P8 is marked by the initial filling of pigment in the center of each ommatidial hexagon (Fig. 3C). This begins from the center of the CE, and gradually spreads toward the periphery so that by the end of the stage the hexagon patterns are no longer recognizable except for those at the eye margin and the CE is uniformly colored (Fig. 3C). The mesosoma now acquires a light pink tinge.

### P9: Black-eyed/Red-thorax pupa

P9 is defined by the completion of eye pigmentation, which is now brown-black, and beginning of pigmentation on other body parts. It also marks the end of the first half of the pupal period (50% of pupation duration). An obvious feature marking the transition to stage P9 is the tanning of a lateral-posterior suture of the mesosoma (Fig. 4). In addition, the red tinge of the mesosoma intensifies, making the color distinct from that of metasoma. Near the end of P9, orange tanning begins in several body parts, including the tarsal claw (Fig. 4), base of mandible, antennal socket, base of proboscis, wing base, tegula, and the suture on the corner of the scutellum. Pepper-colored tanning appears on the pronotum and the margin of the scutellum.

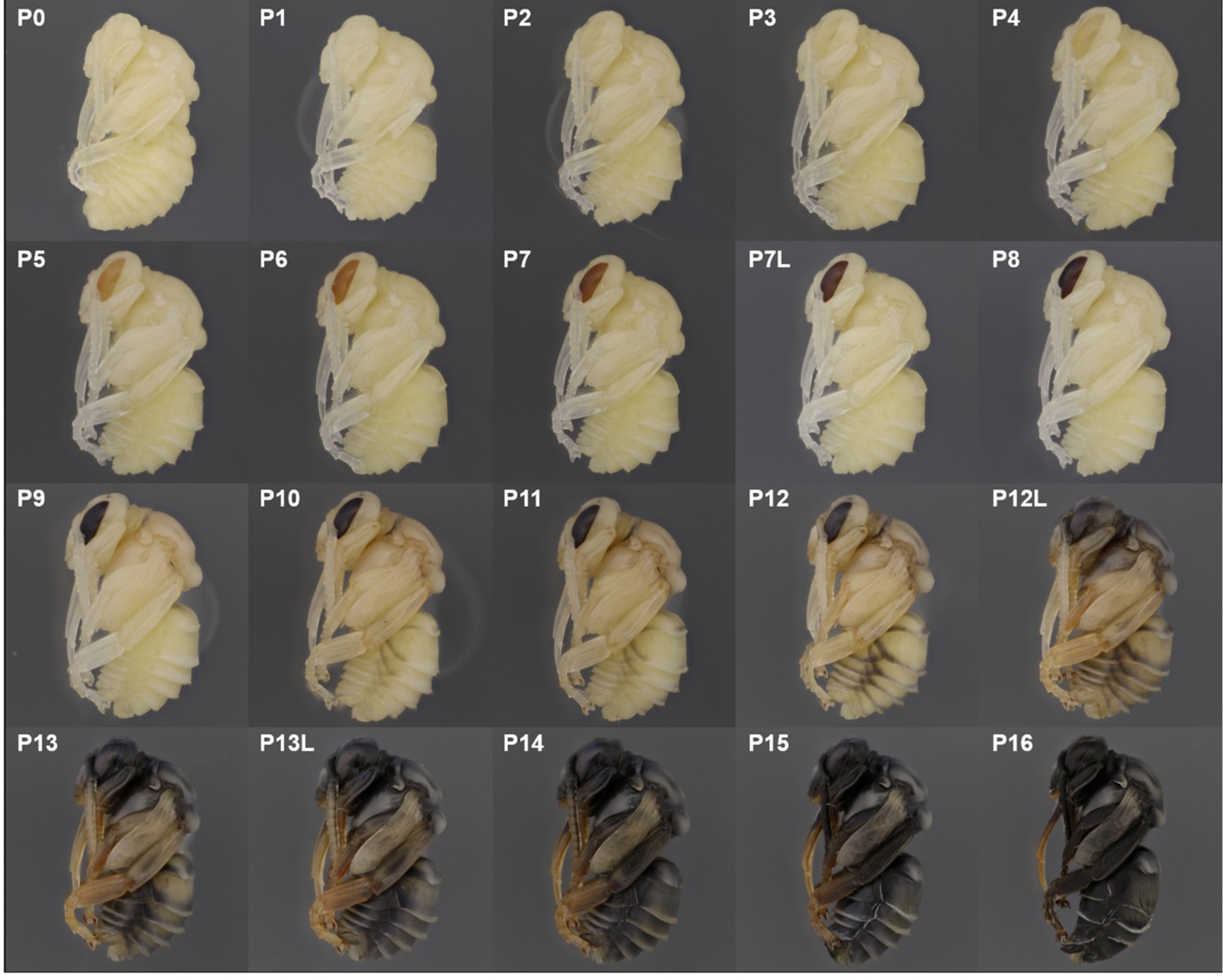

**Figure 7 Photographic guide to developmental stages of *B. impatiens* male pupa.**

### P10: Peppered-A2 pupa

Melanization of the metasoma begins with the apical margin of the first metasomal tergite (Fig. 5) as a lightly peppered stripe. In late P10, stripes appear in the lateral-apical margin of each sternite. Orange tanning of the leg now covers tarsomeres and joints.

### P11: Broken-striped abdomen pupa

Melanization spreads to the apical margin of every metasomal tergite, appearing as a narrow black posterior medial stripe on each white tergite separated from peppering on the lateral side. Peppered tanning begins on the frontal line on the head (Fig. 5) and tanning

begins in other parts as well including the labrum (orange) (Fig. 5), scape of antennae (grayish) (Fig. 5), female sting (orange), basitarsi and tibiae (orange).

### P12/P12L: Zebra pupa

Pepper-colored stripes at the lateral and dorsal apical margins of metasomal tergites are connected to form one continuous stripe circling the entire apical margin (Fig. 5), giving it a zebra-like appearance. The scape turns dark gray (Fig. 5) and the paraocular area and clypeus also turn grayish (Fig. 5). Pigmentation of the black adult setae begins in the late phase of this stage (P12L) (Fig. 5).

### P13/P13L: Appendages blackening pupa

Melanization on metasomal tergites progresses anteriorly. Two other traits marking the transition to P13 include: (1) blackening of basal-lateral proboscis, and (2) antennal darkening that has progressed proximally to now include both scape and pedicel (Fig. 5), with flagellomeres acquiring a gray tinge. Melanization on the head continues, with most of the paraocular area and clypeus now black (Fig. 5). At the late phase of P13 (P13L) melanization covers >1/2 but not all of metasomal tergites and sternites, making the stripe pattern less sharp (Fig. 5). P13L marks the completion of melanization on mesosomal cuticle.

### P14: Black-abdomen pupa

The cuticle of metasomal tergites and sternites in this stage are black (Fig. 5). Future red or black setae begin to melanize, acquiring a grayish-orange tinge, while prospective yellow setae remain white (Fig. 6A). In the case of *B. impatiens,* setal color on A3–A7 (future black setae) is now distinguished from A2 (future yellow setae) (Fig. 6A). This feature is most obvious in queens and large males and workers. In the head, the 1st and 2nd flagellomeres become black, with remaining flagellomeres becoming dark gray (Fig. 5). The paraocular area and clypeus are completely black (Fig. 5) and the mandible is black in the basal 2/3, with the tip reddish. The labrum blackens in the basal 2/3 with reddish color at the margin (Fig. 5).

### P15: Black-bodied pupa

Melanization of all body parts, including antennae, legs, head, and mouthparts finishes, making the non-setal cuticle of the body completely black, with the exception of the distal proboscis and legs which are orange (Fig. 5). The pupal cuticle of the metasoma starts to wrinkle.

### P16: Leg-twitching pupa

Legs start moving freely, perhaps due to the degradation of pupal spines holding the legs in place (Fig. 5). The metasomal segments periscope outward, making the metasoma appear straighter and longer than previous stages. The pupal cuticle is partially degraded in some parts of the body, especially the legs and metasoma, exposing adult cuticle. Pupal cuticle in the head, such as in the labrum and mandibles starts to wrinkle (Fig. 5). This stage, and pupation, ends with molting of the pupal cuticle.

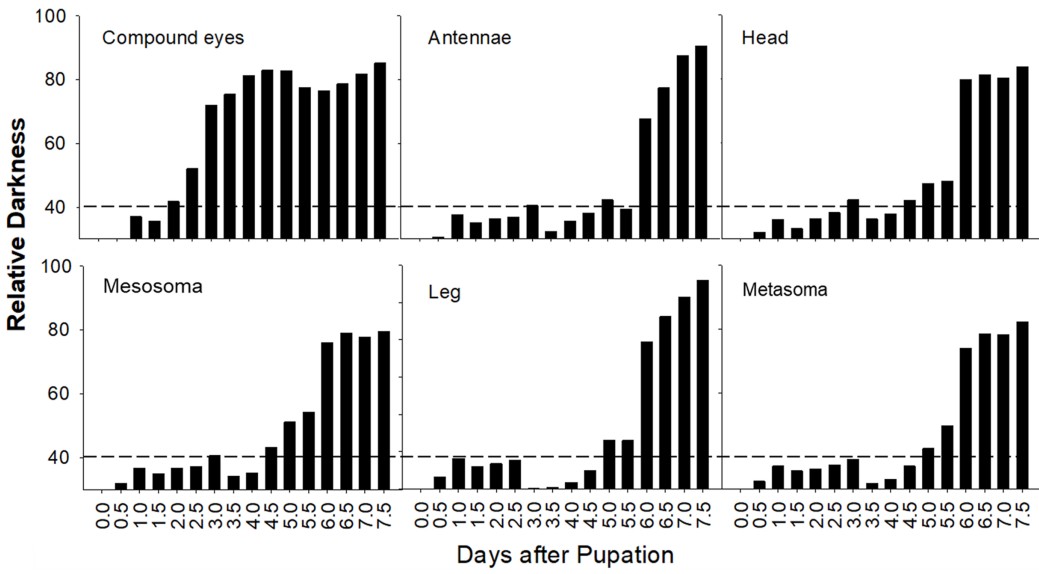

**Figure 8 Variation in timing of melanization among body parts in _B. impatiens_ males.** Relative darkness = gray value of measured area/255 × 100%. The dotted lines indicate background darkness value of the white body, above which the tanning is visible.

### Quiescent adult

This stage is marked by the end of adult ecdysis which takes 30 min. Adult wings spread but are unmelanized and appear transparent. At this stage, the newly eclosed bee remains in the cocoon largely immobile (Fig. 1D). The pupal cuticle has been removed completely, but sometimes is seen attached to the posterior end of the bee. The setae are matted and not fully dried. If removed from the cocoon, the bee will lay on its back, moving its legs, unable to stand or walk.

### Callow

This stage begins when the bee breaks out from cocoon with wings mostly melanized. Callows actively clean themselves and search for food, often hiding and walking slowly among brood as their cuticle finishes sclerotization. These bees are not aggressive and will not sting. This stage covers the first 1–2 days after eclosion and can be considered complete when the bee is fully melanized and behaving like a normal adult in the nest. We examined callows of three species at different ages (hours after eclosion), and found that the primary morphological changes during callow development is setal pigmentation (Fig. 6B).

### General pigmentation patterns

A summary plot for timing of melanization on various body parts is provided in Fig. 8 and the relative progression of CE, cuticle (quantified as the sum of values from Fig. 8) and hair color is summarized in Fig. 9. These plots show very little overlap between eye and body pigmentation. Although pigmentation tends to intensify around the same time across non-eye body parts, the order is somewhat staggered (Fig. 8). Pigmentation is complete first in the mesosoma and head, followed by the metasoma, and then the

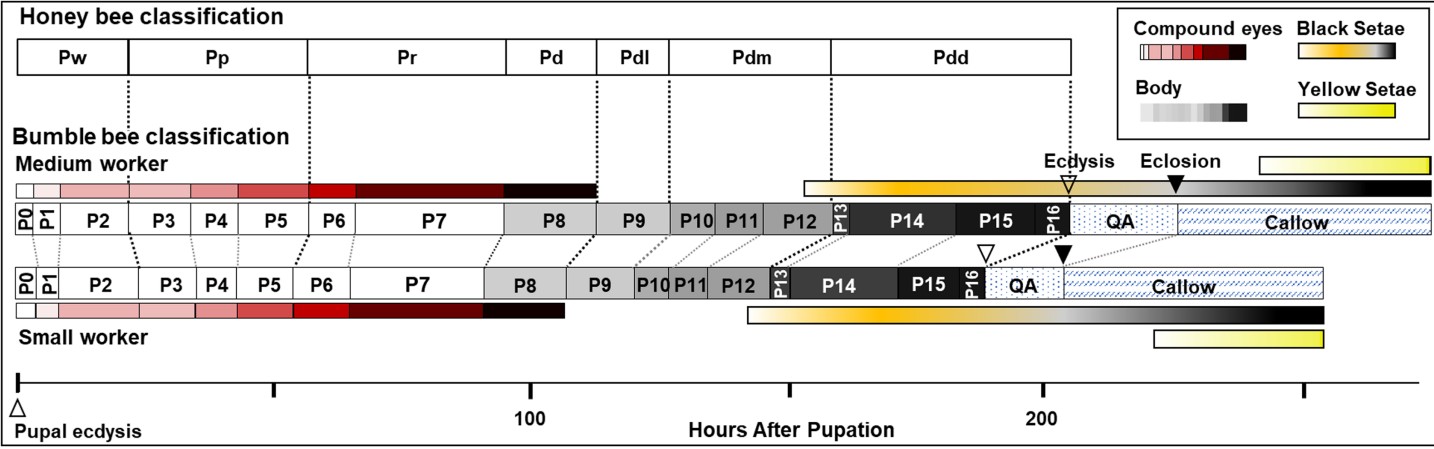

**Figure 9 Summary of stage duration and pigmentation changes in _B. impatiens_ worker pupae and callows.** Colored bars illustrate timing of pigmentation on different body parts. Bumble bee pupal staging applying the honey bee criteria (_Rembold, Kremer & Ulrich, 1980_) is compared to the newly developed staging system. Details on the timing of bumble bee stages relative to the honey bee are provided in Table S2.

appendages, including antennae and legs, which are not fully pigmented until closer to eclosion. Setae follow similar patterns to the main cuticle, although with a slight delay, with prospective black setae starting pigmentation with a light orange tinge after melanization of metasomal cuticle begins and intensifying gradually as pupae age, toward an orange-gray color that is maintained until around the time of adult ecdysis (Fig. 9). At eclosion, prospective black setae are orange-gray, red setae pink-gray, and yellow setae white (Fig. 6B). Color progression of black setae in _B. impatiens_ and _B. vosnesenskii_ and red setae of _B. melanopygus_ begins immediately after eclosion, and is complete about 24–48 h later (Fig. 9). In contrast, pigmentation of yellow setae of all three species examined does not begin until 24 h after eclosion and completes at about 48–72 h (Figs. 6B and 9).

## Pupal duration

Figure 10A compares pupal duration of different phenotype groups reared at 32 and 29 °C. Mean pupal duration and standard error are provided in Table S1. At 32 °C, pupal duration of workers ranges from 160 h (6.6 days, small workers 161.9 ± 1.5 h) to 180 h (7.5 days, large workers 179 ± 1.6 h), males take an average of 189 h (7.9 days, small males 188.6 ± 2.4 h, medium male 189.4 ± 1.7 h), and queens about 243.2 ± 2.2 h (~10 days, Fig. 10A; Table S1). Pupal duration is significantly longer at the lower temperature for all groups (29 °C) (p < 0.01, Mann–Whitney test) (Fig. 10A; Table S1), with variation among groups in the magnitude of response to temperature. In particular, males and queens respond less than workers. Small, medium and large worker pupae reared at 29 °C take 20, 34, and 28 h longer to develop than similarly sized counterparts reared at 32 °C, respectively. However, small males reared at 29 °C take 15 h longer, and queen takes 22 h longer to develop than their 32 °C-reared counterparts (Fig. 10A and Table S1).

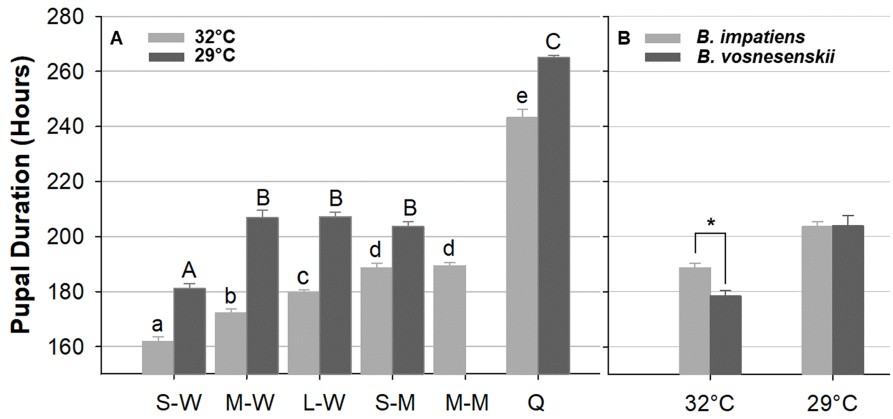

**Figure 10 Bumble bee pupal duration.** (A) *B. impatiens* pupal duration of various castes and body sizes reared at two different temperatures. (B) Pupal duration of *B. impatiens* and *B. vosnesenskii* male pupae reared at different temperatures. S-W, M-W, L-W, small, medium, and large workers, respectively; S-M, M-M, small, and medium males, respectively. Q, queen. Error bars are standard error of the mean (SE). Means with the same letter are not significantly different, One-way ANOVA followed by Tukey's HSD post hoc test, $p < 0.05$; *$p < 0.05$, two-sample *t*-test.       

Overall the ANOVA reveals a strong effect of phenotype (size x caste) on pupal duration (32 °C, $F_{5, 80} = 206.4$, $p < 0.0001$; 29 °C, $F_{4, 29} = 370.12$, $p < 0.0001$, One-Way ANOVA). Body size affects pupal duration in the worker caste significantly, with larger workers taking 18 and 26 h longer to development than small workers at 32 and 29 °C, respectively (Tukey's HSD post hoc test, with α = 0.05) (Fig. 10A). An effect was not found in males of different size (Tukey's HSD post hoc test, with α = 0.05). There were also significant differences between castes at both rearing temperatures (Fig. 10A). In general, queens display substantially longer pupal duration than workers (Fig. 10A). For instance, at 32 °C, queen pupae take 81, 71, and 62 h longer to develop than small, medium, and large sized worker pupae, respectively (Tukey's HSD post hoc test, with α = 0.05). Male pupae also take longer to develop than same-sized worker counterparts (Fig. 10A) with small sized males taking 27 and 22 h longer to develop than small sized worker pupae at 32 and 29 °C, respectively (Tukey's HSD post hoc test, with α = 0.05). The effects of body size within caste also varies between temperatures. Significant differences were found between all three size groups within the worker caste at 32 °C (Tukey's HSD post hoc test, with α = 0.05). However, medium-sized workers were no different from large workers at 29 °C (Tukey's HSD post hoc test, with α = 0.05).

Body size does not alter developmental timing of all stages equally. It appears that the duration of developmental stages for small and medium *B. impatiens* workers were similar up until the orange-eyed pupal stage P5 (first 50 h) (Fig. 9). After this there was an increase in the timing of each stage with increased body size.

Due to limited availability of bees from *B. vosnesenskii* colonies at the time of experiments, only small males at 32 and 29 °C were used for comparison (Fig. 10B). *B. vosnesenskii* male pupae develop 10 h faster than those of similar size in *B. impatiens* at 32 °C ($p = 0.0003$), but no significant difference was found between the two species at 29 °C ($p = 1.0$).

## DISCUSSION

### Morphological transitions in pupal development

The current study provides the first comprehensive description of morphological changes during pupal development in bumble bees. As opposed to previous staging based mostly on eye color, the morphological criteria we established from multiple body parts enables staging of pupal development into smaller, more evenly distributed increments (Fig. 9). This morphological key, annotated with high resolution images, is intended for use in any study on developing bumble bees requiring precise pupal staging.

The characterization of bumble bee pupal stages presented here differs substantially from those described in honey bees and other bee species, with the exception of eye pigmentation (*Elias-Neto, Soares & Bitondi, 2009*). It includes more stages than defined in previous studies on bees, both in eye color and body pigmentation stages (Fig. 9 and Table S2). While the relative timing of cuticle melanization to eye pigmentation is largely consistent with observations in honey bees (*Rembold, Kremer & Ulrich, 1980*), the progression of melanization differs. This is to be expected given the pronounced difference between the species in their external morphology and pigmentation patterns at adult stages, and their large evolutionary distance (>90 my; *Grimaldi & Engel, 2005*). We observed very little difference, however, within and between bumble bee species in pupal traits, consistent with low variability observed within other genera (*Fukutomi et al., 2017*; *Michelette & Soares, 1993*). This key should thus be more broadly applicable across bumble bee species. That said, given that we focused only on a subset of clades within the bumble bee subgenus *Pyrobombus*, studies of non-*Pyrobombus* taxa may want to use this as a guideline for developing a more taxon-specific staging.

Our study highlights several features of pupal development in bumble bees. We first note the features unique to the pupal cuticle that differ from the pharate adult, notably apical tergal spicules and leg protuberances. Large across-species variation has been found in the pupal spicules in bees in terms of their number, size, positions and shapes (*Baker, Kuhn & Bambara, 1985*; *Cardale, 1968*; *Michener, 1954*; *Rozen, 1990*, *2000*; *Torchio & Trostle, 1986*). We noted them to be pronounced in bumble bees but absent in honey bees. Although the function of these is unclear, given their morphology they may serve a purpose as mechanosensory setae or for anchoring while in the pupal cocoon, and thus may differ due to difference in their pupal environment. Pupal leg protuberances, also variably called spines (*Michener, 1954*) or tubercles (*Rozen, 1990*), may be relevant to setal development (*Michener, 1954*) given that they are filled with long setae, but appear to embrace the leg, thus may be holding the legs in place during development. We also observed the shift in pupal segments around the larval-pupal transition. Most non-sawfly Hymenoptera exhibit the synapomorphy of a constriction between their abdominal tergites 1 and 2 that leads to the flexible wasp waist separating the mesosoma from the metasoma. We provide a video (Video S1) observing the regression in these segments that recapitulates the progression of events involved in the formation of the wasp waist.

We have made several observations regarding the shifts in pigmentation across pupal development. We note that the eye follows a dot-to-hexagonal progression likely reflecting development of ommatidia that aligns with shifts in intensification of eye color (Fig. 3C). Eye color also appears to be decoupled from cuticle color, with the cuticle not pigmenting until eye color is complete. Eye color comes from ommochromes used as filtering pigments in the developing ommatidia, a different class of pigments from the brown-black cuticular pigments, which are melanin. Melanization was observed to start in body sutures, including legs joints and sutures on the mesosoma, and then tended to fill in first in the mesosoma, then the metasoma, followed by a progression of darkening from proximal to distal on the appendages. In the metasoma we observed striping patterns in these bees as cuticle melanizes that move posterior to anterior on metasomal tergites (Fig. 5). It is interesting to consider why melanin progresses as it does, with possibilities being related to sources of melanin precursors like tyrosine derived from the center of the body or the availability of oxygen required for melanization (*Lerner & Fitzpatrick, 1950*). Two melanization processes were observed to take place on the primary cuticle, some of which involve gray to black and others which start with a more ochre color, in line with alternative routes to black color in these bees noted by *Milliron (1971)*. This could involve differences in melanin pathways, including differences between pheomelanin and eumelanin or other quinone derivatives (*Hines et al., 2017*). Finally, the yellow setal pigmentation is also fairly decoupled from melanin pigmentation, occurring a day or so after adult eclosion when melanization is mostly complete. This decoupling likely occurs because yellow setae are colored by a different type of pigment, likely a pterin (*Hines, 2008*).

## Pupal duration

We observed shorter pupal duration in all size/caste groups when they were reared at higher temperatures. Both temperatures used are within the typical range of temperatures observed in the brood of bumble bee colonies (*Heinrich, 1972*; *Pomeroy & Plowright, 1980*; *Seeley, 1981*; *Vogt, 1986a*). Higher temperatures within a colony than outside are achieved by the activity of the adults and active thermoregulation of the brood through incubation (*Heinrich, 1972*; *Seeley, 1981*). This suggests that colony development will be impacted by incubation efforts. The negative correlation between rearing temperature and pupal duration is consistent with results from other insects (*Elmes & Wardlaw, 1983*; *Foley, 1981*; *Kipyatkov, Lopatina & Imamgaliev, 2005*; *Penick et al., 2017*). Interestingly, the worker caste is affected more than males and queens by temperature, with greatest effect on medium-sized workers, suggesting that developmentally plastic phenotypes like size and caste may have differential response thresholds to environmental conditions.

Our study reveals a positive correlation between body size and pupal duration within workers, with larger workers taking longer to develop than smaller ones. This could explain variation in worker pupal duration (up to 5 days) in previous studies (*Katayama, 1973*; *Sakagami, Akahira & Zucchi, 1967*). Given that pupal and adult size is determined at the onset of pupation from larval size, pupal duration is likely a consequence of body size rather than vice versa. In our observations, the timing of eye pigmentation phases

did not differ by bee size but later stages involved in melanic pigmentation across the body did (Fig. 9). This may be because the eye is smaller and involves a single-layer (thus reduced complexity) (*Cagan & Ready, 1989*), while melanization would need to take place over a larger surface and may be impacted by signals coming from other parts of the body, which may take longer in larger individuals. The differences we find in both duration and potential lengths of specific stages by body size, means standardization of pupal stages (e.g., for genetic research) is best done on similarly sized individuals.

A positive correlation between body size and duration can fail when sexes and castes are considered. In *Drosophila*, male pupae are smaller than females but take longer to eclose (*Nilssen, 2006*; *Nunney, 2007*), and in the honey bee *A. mellifera*, queen pupae, which are larger than worker pupae, have a shorter pupal duration (*Rembold, Kremer & Ulrich, 1980*). Bumble bee queens do not follow the same patterns as honey bees, as queens have much longer pupal duration than workers and males. This matches expectations from body size, as queens are much larger than workers. The rationale for why a honey bee queen would take less time is interesting to consider. It could be due to their caste being determined more by diet quality (*Kamakura, 2011*). Alternatively, the contrast between bumble bees and honey bees may be explained by their life history: honey bee queens are under strong selection to develop quickly as the first queen to emerge typically becomes the new queen and kills the remaining queen-destined brood (*Degrandi-Hoffman et al., 1998*; *Michener, 1974*), a practice not exhibited by bumble bees which disperse and independently start their own nest. Bumble bee males, however, exhibited longer pupal duration than workers regardless of their body size, and the duration of their development was less sensitive to relative body size. Longer pupal duration in males may be because gonotrophic development is more complicated in males than the non-reproductive worker caste, thus requiring more time to complete metamorphosis (*Breev, Zagretdinov & Minar, 1980*). Different sexes and castes must therefore have adaptations driving duration that are independent of the effects of size.

We also found a difference between *B. impatiens* and *B. vosnesenskii* in pupal duration. Differences within and between closely-related species in developmental timing are not uncommon in insects (*Elmes & Wardlaw, 1983*; *Fukutomi et al., 2017*; *Ma, Huang & Wang, 2015*; *Nilssen, 2006*; *Sgolastra et al., 2012*), and, in bumble bees, differences in habitat have been suggested to select for different developmental rates (*Sakagami, Akahira & Zucchi, 1967*). These two species differ in their climatic niche, as *B. vosnesenskii* is a Pacific coastal bumble bee and has a longer season with milder and wetter weather than *B. impatiens*, which is exposed to eastern North American cool temperate, moderately wet seasonality. This could also explain observed differences in plasticity in these species, as the response to temperature in developmental duration was greater in *B. vosnesenskii* than in *B. impatiens*.

## CONCLUSIONS

This study provides detailed morphological pupal staging criteria, insights into pupal duration under various conditions, and characterization of temporal patterns of pigmentation in bumble bees, all of which should help seed future studies on genetic

and physiological mechanisms regulating adult phenotypes in these developmentally highly plastic and phenotypically diverse bees. Our staging system is established based completely on external morphology of the pupa. It remains to be studied how these features are generated from internal mechanisms and how these stages correspond to critical physiological aspects of the bee, such as hormone titers, ovarian and brain development, that translate to important aspects of phenotypic plasticity in these bees.

## ACKNOWLEDGEMENTS

The authors wish to thank Daniel Snelling, Sarthok Rahman, Elyse McCormick, Istvan Miko and Colin Wright for their constructive comments that helped to improve the manuscript.

### Funding

This work was supported by a National Science Foundation grant NSF DEB #1453473. The funders had no role in study design, data collection and analysis, decision to publish, or preparation of the manuscript.

### Grant Disclosure

The following grant information was disclosed by the authors:
National Science Foundation: NSF DEB #1453473.

### Competing Interests

The authors declare that they have no competing interests.

### Author Contributions

- Li Tian conceived and designed the experiments, performed the experiments, analyzed the data, contributed reagents/materials/analysis tools, prepared figures and/or tables, authored or reviewed drafts of the paper, approved the final draft.
- Heather M. Hines conceived and designed the experiments, contributed reagents/materials/analysis tools, prepared figures and/or tables, authored or reviewed drafts of the paper, approved the final draft.

### Data Availability

The raw data are provided in the Supplementary Files.

### Supplemental Information

Supplemental information for this article can be found online at http://dx.doi.org/10.7717/peerj.6089#supplemental-information.

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
