# Peer review of "Morphological characterization and staging of bumble bee pupae"

_PeerJ, doi:10.7717/peerj.6089_

## Round 0.1 · original submission · Minor Revisions

We received four reviews of your paper which helped make the decision of minor revisions needed. Please address the 3 reviews attached as well as the comments I post here below from another reviewer. I look forward to reviewing your revised manuscript.

Reviewer 4:
-The authors have put a great deal of thought and effort into it. This illustration are well prepared and presented. Other reviewers are probably in a better position to determine whether it has fulfilled its purpose. However, I have a number of problems with it.
-Throughout the manuscript they use the terms “tergite, tergites ” and “sternite, sternites” instead of tergum, terga and sternum, sterna. This is a common errors, but it is an error.
- In addition, they have used the term ecdysis as a noun and adjective, which is correct, but they have also used it as a verb, which it is not, though the meaning is clear.
- there were serious errors in the bibliography in the articles authored by Rozen and Grimaldi. This is hard to understand and suggests that all papers listed should be carefully examined for accuracy of citation.

·

Basic reporting

The authors are native English speakers, so, for me, the English is clear and professional, making easy to understand and read the text. They cited the best references in the area and the Introduction and Discussion proved enough background.

**The figures and schemes present high quality, but some figures need scales.

**The research is a master peace of Bombus pupal development, my unique concern is when the authors compare the development of queens of Apis meliffera with Bombus species. I recommend to eliminate the statement: " The rationale for why a honey bee queen would take less time is/ 557 interesting to consider, and perhaps could be due to their caste being determined more by diet/ 558 quality, differential selection for developmental rate by caste, or alternative physiological /559 mechanisms for generating castes." This is not the objective of the research and the authors can't speculate that.

Experimental design

The research is a master peace of the morphological aspects of the pupal development in Bombus and presents high potential to be used for the scientific community not only for genetic studies, but for toxicological ones as well.
Creative methodology, sufficient detailed, providing enough information for replication.

Validity of the findings

The research is novelty and presents direct impact on many areas, as in molecular regulation of development and developmental toxicology. These bees are serious under threat, not only in USA (B. affinis) as in worldwide. The detailed knowledge of the larval and pupal development of this genus is extremely important for monitoring abnormal conditions and to understand the molecular mechanisms of Bombus development.

Additional comments

Nothing do add.

·

Basic reporting

Robust keys are available in a range of bumble bee species that are often used for studying the morphology of the adult queens, workers, and males. While less is known about the larvae, there is still some literature available on the morphology and development of larvae (with most studies being limited to Bombus terrestris). These have already been used to some degree to study development and changes in gene expression in the larvae of a handful of bumble bee species. However, very little is known about the changes in morphology, and the timing and progression of development in bumble bee pupae, and this study is an attempt to bridge that gap in the literature. In my view the authors have done this very successfully, and as such I would like to recommend that this article be accepted for publication, with the caveat that they consider a few relatively minor revisions.

In general, the written English is of a high standard, and was very easy to follow throughout. However, there were one or two minor grammatical mistakes that the authors should correct before their final submission. I noticed a few sections where the authors confuse tenses, for example on line 86 the sentence ‘…pre-pupa shed the larval cuticle…’ should read ‘…pre-pupa sheds the larval cuticle….’ While this may seem like a very minor issue, I noticed several instances where the authors make similar mistakes (other examples include lines 110, where ‘is’ should be replaced with ‘are’, line 126 which should read ‘…develop for four days…’, line 246, which should read ‘…on the coxae and trochanters…’, line 434, which should read ‘…there are also significant differences between…’, line 537 should read ‘… larger workers taking longer to develop than smaller ones...’, line 564 should read ‘that override the effects of size’) I would therefore advise a very careful rereading of the text to remove any similar errors.

In addition, they may need to clarify one or two sections, for example on line 151 I was unclear what they meant by ‘the same source of fresh pollen’. If they meant that the pollen arrived with the colony, or that they bought the pollen from Koppert, then they should state this more clearly.

The authors provide an extensive list of citations, and mostly cite the relevant literature in the field, however they very occasionally make points without providing relevant examples. Therefore I would like to see a few further citations to back up their points on lines 73 (sentence starting ‘For example,’), 207 (Sentence starting ‘Previous studies…’, those studies need to be cited) and 516 (sentence starting ‘Many studies…’ these studies should also be cited). They should recheck their introduction and discussion to ensure there are no other statements made that would usually require citations.
The structure of the article is fine, and follows a logical order. While they have a large number of figures, and a large proportion of them are photos, these represent the primary data, and are quite usual for a study of this kind. I would encourage the authors to ensure the figures are presented in the correct order because currently figure 7 is first described on line 266 whereas figure 6 is not described until line 367. I suggest that the authors check for any other discrepancies like this and correct them.

On lines 422-429, the authors describe the ranges of pupal durations for each of the described castes. While they state that the means and standard errors are provided on table S1, I would prefer them to also include the means and standard errors for each of the values described on line 424. Furthermore, they need to ensure that they are consistent in the units they use. For example, in the text in this section they refer to the timings of pupal duration in days, whereas in table S1 they use a different metric (I presume they use hours in the table, but it does not state this, so that also needs to be corrected). Some of their statistical reporting could be improved in this section too, for example on line 426 they say that pupal duration is significantly longer at the lower temperature of 29, but they don’t describe the magnitude of the effect (or range of effects). I didn’t find the results presented on lines 427-429 very clear so I think they should re-write that section, expand on any of the results presented if necessary, and should reference the appropriate statistical tests in support of each of their conclusions. Likewise for lines 430-439, some of the results presented don’t reference the appropriate statistical justification, or the figures or the tables, and the authors need to ensure that the language used should present these results as clearly as possible.

Finally, the sentence starting on line 563 did not make sense to me and I think it needs to be re-written to make their point more clear. I should also say that despite the inference that body size is ‘driving’ development time, I think a plausible argument could easily be the converse of this, i.e. that development time is actually driving changes in body size, or that a third factor is driving the changes in those two variables.

Experimental design

The methods section was generally highly detailed, and I felt like they provided sufficient information for the study to be easily replicated. My understanding of the methods is that they cut open the cocoons, then follow the developmental progress of all the individual that were found to be either silk-spinning larvae or pre-pupae (Line 155-163). However, it is not clear what (if anything) they do when they first open cocoons that already contain pupae (including late-stage pupae). My guess is that these were ignored or discarded, but this needs to be stated clearly, together with the sample sizes which include the numbers of cocoons opened, the numbers that were monitored from larvae/pre-pupae and the numbers that already contained pupae. On the other hand, if they used the cocoons that already contained pupae for the experiment, then they need to provide more detail about the sample sizes of pupae that were monitored from different starting points, and how they were able to distinguish pupae that were different ages.

In the sentence starting ‘Based on measurement…’ on line 221, the authors categorize the adult workers and males as small, medium, or large depending on the ITS. It’s not clear whether those size ranges mean anything biologically, or were based on the mean and ranges of the adults that they sampled, or were just arbitrarily chosen. It would be helpful if they could state this, and/or provide literature citations to back up why they classified adults into those particular size ranges. This is important because many of their results and conclusions were dependent on the sizes of the adult individuals that they were measuring. In addition, my guess is that they determined those size categories retroactively, once the pupae they were monitoring had eclosed as adults. Once they had eclosed, the authors then analysed data separately for pupae that were known to develop into different sized males and workers. If they did categorize them this way then I would like them to directly state this (or otherwise make it clearer if I have interpreted this wrongly).

Validity of the findings

Overall, I felt like the data were robust, and will inevitably be of wider use to the field in general. They were able to state their main conclusions, and these were well supported by their results. During some sections they speculate where their results might be more broadly applicable to researchers studying other bumble bee taxa, but are clear where their data might have limitations, for example on line 473 where they state that these data might only be applicable to Pyrobombus species.

I found the section from lines 515-520 to be unnecessary. They state (without citation) that many studies use the production of, and eclosion from, cocoons to define pupation and that this is a problem because pupation doesn’t occur until sometime after the cocoons are made, and ends some time before the new adults eclose. However, the authors do not provide any new data to show when the precise timing of pupation should be defined, and do not explicitly state any firm conclusions as to what future studies should do to avoid the issues they claim are problematic. For these reasons I would recommend they simply remove that paragraph.

Additional comments

Overall I thought the study was good, and assuming the authors consider the points and suggestions that I have raised, it should easily meet the criteria for publication in PeerJ.

I had a brief comment about one of findings of the study. On line 556 the authors speculate on the reasons why larger bodied honeybee queens develop more quickly than workers, whereas in bumble bees they found that it was the other way around. One likely reason for this, which they elude to but don’t explicitly state, is that honey bee queens are under immense selection pressure to develop quickly, because the first queen to emerge will usually take-over the colony, and then kill any rival queen-destined larvae and pupae while they are still developing in their cells. There is no analogous process to this in bumble bees, as queens do not take-over the colony, and simply disperse once they have eclosed. Therefore, there is no reason why bumble bee queens should develop more quickly than workers which is why bumble bees have a more simple linear relationship between body size and development time compared to honey bees.

·

Basic reporting

Tian and Hines investigated morphological changes during pupal development of bumble bees and provided criteria for staging. The English language is clear and professional. The manuscript is very well structured, including high quality supporting files (figures, tables and so on). Cited references are OK.

Experimental design

The study is original because it improves the current staging criteria applied for Bombus pupae using more morphological details, thus filling some scientific gaps. The work is technically and ethically appropriate. The methodology and experiments are well described and designed, respectively.

Validity of the findings

Results, statistics, discussions and conclusions are relevant and robust.

Additional comments

Minor suggestions:

- Change title to: Staging of bumble bee pupae based on morphological criteria

- Replace Bumble bees (Hymenoptera: Bombus) by Bumble bees (Hymenoptera: Bombinae)

- Replace "de F Michelette E, and Soares A" by "Michelette EF, and Soares AEE"

---

## Round 0.2 · accepted · Accept

Thank you for addressing the reviewer comments in detail. Congratulations on the acceptance of your manuscript.

#